# Transcriptomic Sequencing Analysis on Key Genes and Pathways Regulating Cadmium (Cd) in Ryegrass (*Lolium perenne* L.) under Different Cadmium Concentrations

**DOI:** 10.3390/toxics10120734

**Published:** 2022-11-28

**Authors:** Bingjian Cui, Chuncheng Liu, Chao Hu, Shengxian Liang

**Affiliations:** 1Institute of Farmland Irrigation, Chinese Academy of Agricultural Sciences, Xinxiang 453002, China; 2Key Laboratory of High-Efficient and Safe Utilization of Agriculture Water Resources, Chinese Academy of Agricultural Sciences, Xinxiang 453002, China; 3Institute of Life Sciences and Green Development, College of Life Sciences, Hebei University, Baoding 071000, China

**Keywords:** perennial ryegrass, cadmium, RNA-Seq, RT-qPCR

## Abstract

Perennial ryegrass (*Lolium perenne* L.) is an important forage grass and has the potential to be used in phytoremediation, while little information is available regarding the transcriptome profiling of ryegrass leaves in response to high levels of Cd. To investigate and uncover the physiological responses and gene expression characteristics of perennial ryegrass under Cd stress, a pot experiment was performed to study the transcriptomic profiles of ryegrass with Cd-spiked soils. Transcriptome sequencing and comparative analysis were performed on the Illumina RNA-Seq platform at different concentrations of Cd-treated (0, 50 and 500 mg·kg^−1^ soil) ryegrass leaves and differentially expressed genes (DEGs) were verified by RT-qPCR. The results show that high concentrations of Cd significantly inhibited the growth of ryegrass, while the lower concentrations (5 and 25 mg·kg^−1^) showed minor effects. The activity levels of antioxidant enzymes such as superoxide dismutase (SOD), peroxidase (POD), catalase (CAT) and malondialdehyde (MDA) increased in Cd-treated ryegrass leaves. We identified 1103 differentially expressed genes (DEGs) and profiled the molecular regulatory pathways of ryegrass leaves with Gene Ontology (GO) and Kyoto Encyclopedia of Genes and Genomes (KEGG) analysis in response to Cd stress. Cd stress significantly increased the membrane part, the metabolic process, the cellular process and catalytic activity. The numbers of unigenes related to signal transduction mechanisms, post-translational modification, replication, recombination and repair significantly increased. KEGG function annotation and enrichment analysis were performed based on DEGs with different treatments, indicating that the MAPK signaling pathway, the mRNA surveillance pathway and RNA transport were regulated significantly. Taken together, this study explores the effect of Cd stress on the growth physiology and gene level of ryegrass, thus highlighting significance of preventing and controlling heavy metal pollution in the future.

## 1. Introduction

With the development of agro-industrial production, the improper management of heavy metal-containing wastewater, sludge and fertilizer in agricultural practice is responsible for an increase in the soil’s heavy metal content. Cadmium (Cd) is one of the most important and poisonous heavy metal pollution elements, which is very harmful to plants and the human body. Cd can be readily absorbed by the plant roots and migrated to grains or fruits, and enter into the human body through the food chain, thus posing a threat to public and ecosystem health [1,2].

As a kind of green remediation biotechnology, phytoremediation involves the use of plants to uptake and accumulate heavy metals from contaminated soil or water [3]. Plants accumulate and precipitate heavy metals through extraction, stabilization, immobilization, volatilization and filtration, and can still grow normally without being inhibited [4,5,6]. Plants are known to have different capacities to accumulate heavy metals, and their potential to accumulate heavy metals mainly depends on their tolerance ability and biomass [7]. The selected plants with high biomass have a higher accumulation ability, while those with a low biomass and slow growth of plant species are less effective. Ryegrass (*Lolium perenne* L.) can accumulate and concentrate relatively high levels of cadmium (Cd). A previous study has suggested that perennial ryegrass can grow normally on heavy metal-polluted soil, which is attributed to its higher tolerance [8]. Therefore, perennial ryegrass with a high biomass is a suitable variety for phytoremediate Cd-contaminated soil. The Cd-treated perennial ryegrass exhibited increased malondialdehyde levels and antioxidant enzyme activity [9]. qRT-PCR, used for determining changes in gene expression levels, is essential in order to select one or more appropriate reference genes for normalization, as several stably expressed genes (*eEF1A*, *YT521-B* and *eIF4A*) have been validated as reference genes for perennial ryegrass [10]. The expression stability of reference genes in perennial ryegrass under various stresses has been comprehensively studied. eIF4A, TEF1, E2 and TBP-1 exhibited the most stable expression, and were recommended as suitable reference genes for various forms of abiotic stresses study in perennial ryegrass [11]. Transcriptome analysis has been widely used in different plant species to effectively identify gene expression under different abiotic and biotic stresses. A wide range of salinity-regulated genes has been associated with the salinity tolerance of perennial ryegrass, which could be categorized into 11 different functional groups and used as candidate genes to understand plant-adaptive mechanisms to salinity stress [12]. An RNA-Seq strategy has developed a transcriptome reference for perennial ryegrass, the assembly of which has greatly enriched the public databases for perennial ryegrass RNA sequences that facilitate the determination of genetic variation, expression analysis, genome annotation and gene mapping [13]. A comparative transcriptomic study under Cd stress revealed several Cd-related transporters involved in the uptake, tolerance and detoxification of Indian mustard (*Brassica juncea*) [14]. A Kyoto Encyclopedia of Genes and Genomes (KEGG) analysis revealed that several metabolic pathways in wheat (*Triticum urartu*) were induced under Cd stress, especially DNA replication and phenylpropanoid biosynthesis [15]. Most of the differentially expressed genes (DEGs) identified in perennial ryegrass under heat stress were relatively common to the genes reported to be responsive to heat stress in plants, including HSFs, HSPs and antioxidant-related genes [16]. It was found that the tolerance of ryegrass to Cd could be improved by increasing the expression levels of MT family genes and Nramp2 genes, but the two gene families were not associated with Cd transport from root to leaf or Cd accumulation in the above-ground parts [17]. Hu et al. (2020) provided a comprehensive draft of the regulatory networks in Italian ryegrass root under Cd stress, and characterized a DEG LmAUX1 expressed in *Arabidopsis thaliana* significantly enhanced by plant Cd accumulation [18].

Although previous studies have demonstrated that perennial ryegrass has the potential to remedy Cd-contaminated soils, further studies need to be carried out to better understand Cd-induced gene expression and metabolic pathways in perennial ryegrass. Therefore, it is paramount to identify the expression of key genes that elucidate the mechanism of Cd tolerance in ryegrass under low or high levels of Cd pollution. In this study, we determined the physiological characteristics and gene expression of perennial ryegrass under Cd stress, and investigated the transcriptional response to Cd stress in leaves of perennial ryegrass using transcriptomic analysis. The results strengthen and improve our understanding of the gene expression variability at the transcriptome level by providing valuable and useful information about the metabolic signaling pathway and molecular mechanisms of perennial ryegrass under Cd stress.

## 2. Results

### 2.1. Effects of Cd Exposure on Ryegrass Growth and the Physiological Properties

Ryegrass showed different growth characteristics that responded remarkably to different Cd levels (Appendix A). The increase in Cd concentration caused the plant height, fresh weight and tiller number of ryegrass to decrease (Figure 1a–c). The content of chlorophyll a was not significantly different compared with the control without spiked Cd, while the soluble protein in ryegrass increased first and then decreased (Figure 1d,e). The content of Cd in roots and leaves of ryegrass increased significantly with the increase in Cd-spiked soil concentration, and reached a peak value at 500 mg·kg^–1^ (Appendix A). The Cd content in roots of ryegrass was higher than that in leaves under an identical Cd concentration, indicating that most of the Cd absorbed by ryegrass was mainly retained in roots (Appendix A). The Cd translocation factor (TF) of ryegrass was less than 1.0. Compared with the control, the bioconcentration factor (BCF) in ryegrass leaves was less than 1.0, while the roots only demonstrated a BCF less than 1.0 from 500 mg·kg^–1^ of the Cd-spiked soil (Appendix A).

### 2.2. Responses of Antioxidant Enzymes of Ryegrass under Cd Stress

Cd significantly affects the activity of antioxidant enzymes in plants (Appendix A). In a certain range, the activities of antioxidant enzymes increased with an increase in the Cd concentration. When the concentration of Cd was high enough, it decreased the activity of antioxidant enzymes in ryegrass leaves. Compared with the control, the activity of SOD, POD and CAT in leaves induced a significant increase following Cd treatment, as shown in Figure 2. When the concentration of Cd reached 500 mg·kg^–1^, SOD activity decreased significantly (*p* < 0.05) (Figure 2b), while the activity of CAT and POD only slightly decreased (Figure 2c,d), but with no significant difference. This indicates that ryegrass had a certain tolerance range to Cd toxicity. The degree of membrane lipid peroxidation and the content of malondialdehyde (MDA) were significantly increased under high concentrations of Cd stress (*p* < 0.05) (Figure 2a).

Heavy metal stress resulted in the production and accumulation of oxygen-free radicals in plant cells, which significantly affected the activities and gene expression of antioxidant enzymes (Appendix A). The relative expression of antioxidant enzyme genes was evaluated for ryegrass leaves in different Cd-spiking treatments, Figure 3 shows the magnitude of these expression changes. An increase in *POD* expression was observed in the leaves under C25 treatment, while there was no significant change in other treatments (Figure 3a). The gene expression of *CAT* changed as the heavy metal concentration increased (Figure 3b). Compared with the treatment without heavy metal, *APX* and *Cu/ZnSOD* expressions in ryegrass leaves were significantly increased under C50 and C500 treatment (*p* < 0.05) (Figure 3c,d). The gene expression of *FeSOD* was gradually assisted with increasing Cd concentration (Figure 3e). No significant differences in the levels of *MnSOD* expression were observed in ryegrass leaves under all treatments, except for C100 (Figure 3f). The gene expression of antioxidant enzymes related to antioxidant metabolism was affected by Cd stress, while the expression level of antioxidant enzyme genes indicated the adaptability of ryegrass to Cd stress.

### 2.3. Transcriptional Changes of Ryegrass under Cd Stress

Comparative transcriptomic analysis was performed on the Illumina RNA-Seq platform at different concentrations of Cd-treated (0, 50 and 500 mg·kg^–1^ soil) ryegrasses. In total, 62.53 Gb of clean data were obtained by the transcriptomic analysis of nine samples, and the percentage of Q30 base in each sample was more than 94.47% (Appendix A). After removing low-quality reads, the transcriptome was assembled into 118,443 unigenes, producing an average length of 754 bp. A detailed summary of the Illumina transcriptome sequencing is provided in Appendix A. The lengths of all the assembled unigenes and transcripts were calculated, and most of the assembled sequences 56% (unigenes) and 48% (transcripts) sequences in each sample were less than 500 bp, respectively (Appendix A). The clean reads were mapped to the assembled unigenes, and the mapped ratio > 71% could map to the unigenes (Appendix A).

After the assembly, annotations for the assembled unigenes were performed by BLAST in six public databases (NR, Swiss-Prot, Pfam, COG, GO and KEGG) with a cut-off E-value of 1.0 × 10^–5^. In total, 52,373, 32,703, 32,907, 48,414, 43,497 and 20,654 unigenes were aligned, respectively (Appendix A). Additionally, 53,550 unigenes were annotated in the database, accounting for 45.21% of all unigenes. The unigene assemblage was annotated with NR, and the proportion of species in the assemblage was as follows: *Triticum urartu* (4589, 23.66%); *Triticum aestivum* (3031, 15.54%); *Oryza sativa* (2446, 12.61%); and *Zea mays* (1010, 5.21%), respectively.

Principal component analysis (PCA) was performed based on the expression levels of all genes to identify the overall variance in the transcript data. The results indicate that the gene expression in C0, C50 and C500 groups were obviously distinguished, and the first two principal components (PC1 and PC2) accounted for 14.66% and13.48% of the total variance, respectively (Appendix A). A Venn diagram was generated from the expressed unigenes to illustrate the number of shared and specific unigenes among all samples. The leaf-specific expression genes were calculated to reveal differences in the unigenes of various ryegrass leaves. In total, 118,443 unigenes were detected, of which 11,257 were shared among the different Cd stress treatments. Additionally, 1793, 1481 and 1374 unigenes were specifically expressed in C0, C50 and C500, respectively (Appendix A).

To determine genes that exhibited significant variations in expression under Cd stress, differentially expressed genes (DEGs) were dissected by comparing the control and treated libraries. DEG analysis identified 302 (149 up- and 153 down-regulated) and 452 (226 up- and 226 down-regulated) genes expressed differentially for the C0_vs_C50 and C0_vs_C500 treatments, respectively (*p*-adjust < 0.05, |log_2_FC| ≥ 1) (Figure 4a). The two groups shared 62 DEGs in common, and 240 and 390 unique DEGs were presented among the two groups (Figure 4b).

In total, 1103 genes were differentially expressed among Cd treatments, and a heatmap clustering analysis of all DEGs showed that the untreated and Cd-treated samples were clearly separated (Appendix A). Here, we reported the transcriptome information and metabolic changes in ryegrass leaves under different stress concentrations, and classified and analyzed them according to their metabolic pathways or gene functions. GO enrichment and KEGG pathway enrichment analysis were used to identify the important biological processes, cellular components and molecular function in ryegrass leaves in response to different concentrations of Cd. The GO functional analysis showed that the specifically expressed genes under Cd stress were completely assigned into three categories: molecular functions (MFs, 11); biological processes (BPs, 13); and cellular component (CCs, 11) (Figure 5). Among the MFs, the DEGs in three groups were mainly annotated into the binding (GO:0005488), catalytic activity (GO:0003824), transporter activity (GO:0005215), antioxidant activity (GO:0016209) and transcription regulator activity (GO:0140110) categories. Biological regulation (GO:0065007), metabolic processes (GO:0008152), cellular processes (GO:0009987) and response to stimuli (GO:0050896) were the most abundant functional terms in BP. In the CC category, the cell part (GO:0044464), membrane part (GO:0044425), membrane (GO:0016020), protein-containing complex (GO:0032991), organelle (GO:0043226) and organelle part (GO:0044422) were the six most-represented terms. Furthermore, these genes have the highest proportion of functional annotations in C500. The GO enrichment analysis of common DEGs showed that specific genes were mainly enriched in transferase activity, response to stress, response to oxidative stress, peroxidase activity and other functions. KEGG signal pathway annotation classification results showed that 1290 genes were involved in cellular process, 4435 genes in genetic information processing, 1062 genes in environmental information processing and 7942 genes in metabolism. In addition, to further illustrate the pathways associated with the specifically expressed genes, the top 20 KEGG pathways were classified into five metabolic groups (Appendix A). The KEGG pathways were consistently classified and the number of genes in C500 group was significantly higher than that in the C50 group. Many genes were associated with metabolism and organismal systems in C500, while C500 involved metabolism, environmental information processing, genetic information processing and organismal systems. A bubble chart of the represented genes displays the top 20 most-enriched pathways (Appendix A). The degree of KEGG enrichment was evaluated by the rich factor, FDR and gene number. Several significantly enriched pathways were involved in glycerophospholipid metabolism, cysteine and methionine metabolism, RNA transport and MAPK signaling pathway-plant.

### 2.4. Validation of DEGs

To further verify the genes related to the internal regulation of ryegrass leaves under Cd stress, FDR ≤ 0.001 and FC (Fold Change) ≥ 2 were used as the screening criteria to pick out 10 DEGs with significant regulation under Cd stress. The qRT-PCR was carried out for these selected genes with different sets of primers to assess the relative expression levels in response to Cd stress (Figure 6). These DEGs included five up-regulated and five down-regulated genes, which were consistent with the results of transcriptome sequencing. The classification of the DEGs, according to GO annotation, showed that the genes were involved in the biological processes of S-adenosylmethioninamine biosynthetic process, the stress-activated protein kinase signaling cascade, the defense response and the carbohydrate metabolic process; the cellular components of the integral component of membrane, cytosol and cytoplasm; and the molecular functions of adenosylmethionine decarboxylase activity, calcium ion binding, metal ion binding and protein kinase activity.

## 3. Discussion

Phytoremediation is a promising green technology used to treat environmental pollution, and the remediation effect depends on the tolerance, migration and bioavailability of plants. Cd is a non-essential biological element and harmful element of environmental pollution, which can cause toxic effects on organisms. Ryegrass is an important perennial forage grass that can tolerate and accumulate heavy metals, and many studies have reported that it has strong tolerance and enrichment abilities for a set of heavy metals, including cadmium [19,20,21]. It can be seen that cadmium stress has a great effect on the physiological and biochemical properties of ryegrass leaves. When plants absorb or accumulate a certain amount of heavy metals, they will produce toxic effects such as inhibited growth development, physiological disorder, blocked various metabolic activities and decreased yield [22,23]. Apart from chlorophyll and soluble proteins, ryegrass leaves significantly responded to a high concentration of Cd. It was found that Cd stress could trigger antioxidant enzyme activity and increase the lipid peroxidation levels of ryegrass [24]. Plants under environmental stress are conducive to the production of reactive oxygen species, resulting in membrane lipid peroxidation and abnormal plant growth. Plants exposed to 500 mg·kg^−1^ of Cd had reduced stem height, weight and tillers of ryegrass, and also resulted in oxidative stress, as indicated by lipid peroxidation and oxidant enzyme activities. The content of Cd in ryegrass treated with different concentrations of Cd was determined, and it was found that the enrichment degree of Cd in ryegrass was in the order of root > leaf. Bioconcentration and translocation factors (BCFs) confirmed that Cd was preferentially accumulated in the root, which was in line with that of Bidar et al. (2009) [25]. The BCF and TF of Cd significantly decreased with the increase in Cd concentrations. The BCF was > 1 for roots and < 1 for leaves. Plants possess various aptitudes to alleviate Cd stress, including reactive oxygen species (ROS) scavenging and chelation-mediated Cd detoxification [24].

The effect of heavy metal stress on the expression of genes encoding antioxidant enzymes in ryegrass leaves was studied. SOD, POD and CAT are important enzymes in plant adaptation to environmental stress, and Cd stress enhanced the antioxidant stress and free-radical scavenging ability of ryegrass. In a certain range of Cd concentration, the antioxidant enzyme activity of plants was maintained or increased, and the activity decreased when the Cd concentration exceeded the threshold value. Moreover, 50 mM of CdCl_2_ resulted in an increased MDA content and reduced activities of SOD and CAT in leaves of four wheat (*Triticum aestivum*) lines [26]. Notably, 500 mg·kg^−1^ of Cd exposure significantly deregulated the physiology of ryegrass in this study. The gene expression of antioxidant enzymes in ryegrass leaves was affected by various Cd stress. Luo et al. (2011) indicated that antioxidant enzyme activity may play an important role in improving heavy metal tolerance, mainly due to the stable or increased gene expression of antioxidant enzymes that may confer Cd tolerance on perennial ryegrass [9]. The plant could adapt to heavy metal stress by regulating gene expression, as Cu- and Zn-SOD gene expression was induced by exposure to Cd stress on the level of mRNA accumulation in the roots of soybean (*Glycine max*) [27]. The gene expression levels of SOD, POD, CAT and APX were induced by Cd, and progressively increased when exposed to high concentrations of Cd. Plants usually activate antioxidant defenses and alter cellular metabolism to mitigate the oxidative damage initiated by heavy metals [28,29,30]. The decrease is related to the role of the corresponding antioxidant enzyme system encoded by these genes in anti-stress metabolism. The expression of related genes in plants responded positively to the stress adaptation of plant metabolism.

Heavy metals have harmful effects on plants, and there are many tolerance mechanisms for cadmium stress in plants, so the accumulation, metabolism and destruction of plant cells need to be considered. The plant chosen was perennial ryegrass, which is known for its high biomass and resistance to heavy metals stress or toxicity [31]. So far, only a draft genome and NAC transcription factor family of perennial ryegrass have been reported [32,33]. The RNA-Seq strategy was applied to non-model species, and Farrell et al. (2013) reported de novo transcriptome assemblies for perennial ryegrass that could help to determine expression analysis, genome annotation and gene mapping [13]. Metallothioneins (MTs) are small metal-binding proteins that can bind with heavy metal ions to form non-toxic or low-toxic compounds; they play various roles in plant abiotic stresses such as metal detoxification, nutrient remobilization, ROS scavenging, tolerance, plant development and cellular metal ion homeostasis [34]. The natural resistance-associated macrophage protein (NRAMP) is part of a membrane-integrated transport protein family, which can transport metal ions such as Mn^2+^, Zn^2+^, Cu^2+^, Fe^2+^, Cd^2+^, Ni^2+^ and Co^2+^. The expression amount of MT and NRAMP family genes in leaves was higher than that in the root and stem, suggesting that the two gene families might be closely related to Cd uptake and tolerance [17]. Phytochelatins (PCs) can chelate Cd^2+^, Zn^2+^ and Cu^2+^ by entering plant cells, thus alleviating the toxicity of heavy metals on plant cells [35]. Moreover, PCs also play a major role in metal ion homeostasis, and then regulate the metal ion availability in plant cells [36]. MTs and PCs may act as cellular homeostatic or detoxifying agents; they are likely to interact with plant antioxidant defense systems or become involved in translocating and distributing excessive ion metals [37]. Domínguez-Solís et al. (2012) found that O-acetylserinethiolase (OASTL) could promote the increase in PC content and improve Cd tolerance in plants [38]. A study has shown that OAS and IRT could regulate cadmium transport and uptake in ryegrass [39]. Our findings suggest that transcript profiling analysis provides deep insights into the integrated molecular mechanism of Cd tolerance in ryegrass leaves.

Comparative transcriptomic analysis was used to identify metabolic regulation and response genes of ryegrass leaves under Cd stress. Plenty of key genes have been demonstrated to be differently induced or repressed differently in ryegrass leaves under Cd stress. Cd toxicity induces large-scale changes in gene expression. Compared with the control, the gene expression profiles changed significantly after exposure to 50 and 500 mg·kg^−1^ of Cd stress. In total, 302 and 452 genes were differentially expressed between Cd-stressed and control plants, respectively. Of these genes, 149 and 153 genes were up- and down-regulated in C0_vs_C50, while there were more genes upregulated and down-regulated in C0_vs_C500. The selected DEGs were verified by RT-qPCR, and the results are consistent with the results of transcriptome analysis, confirming the accuracy of transcriptome sequencing results. Plant cells produce different defense mechanisms by altering gene expression patterns to reduce the effects of abiotic or biotic stresses. Genes encoding fructan 6G-fructosyltransferase (DN53494_c0_g1), ligase (DN5585_c0_g1) and epoxidase (DN92214_c0_g2) were up-regulated under Cd stress, while chitinase (DN75_c0_g1), kinase (DN4816_c0_g1) and phosphatase (DN18177_c0_g1) were down-regulated. The GO enrichment analysis of DEGs revealed that they were mainly classified into the biological processes (BPs), cell components (CCs) and molecular functions (MFs). The biological process is usually accomplished by a particular set of molecular functions carried out by specific gene products (or macromolecular complexes). Response to stress (GO:0006950), DNA metabolic processes (GO:0006259), DNA integration (GO:0015074) and response to oxidative stress (GO:0006979) account for a higher proportion of biological processes. Molecular functions can be carried out by the action of a single macromolecular machine, and these actions are described from two distinct but related perspectives: biochemical activity; and their role as a component in a larger system/process. Molecular function is mainly enriched by antioxidant activity (GO:0016209), peptidase activity (GO:0008233), protein serine/threonine kinase activity (GO:0004674) and transferase activity (GO:0016772). In addition to these differentially regulated genes involved in stress and defense responses, there were also membrane region (GO:0098589) and membrane-bounded vesicles (GO:0031988) belonging to the cellular component where the gene product performs its function. The regulation of Cd stress in ryegrass involved biosynthesis, signal transduction and metabolisms, which is helpful to elucidate the molecular mechanism of ryegrass responses to high-concentration Cd stress. Comparative transcriptome analysis revealed key genes associated with Cd tolerance in barley leaves, including those encoding proteins related to stress and defense responses and metabolism-related genes involved in detoxification pathways [40]. Ryegrass is a promising plant with a significant ability to remediate heavy metals, and the presence of tolerance mechanisms can be observed at both the morphological development and molecular levels. Plant tolerance to heavy metal stress was counterbalanced by detoxification mechanisms. The study found that the expression of genes that regulate antioxidant activity and stress increased when the plant was exposed to heavy metals [41].

## 4. Conclusions

In this study, the transcriptomic and physiological analysis revealed the cadmium tolerance mechanisms of *Lolium perenne* L. Ryegrass showed physiological and biochemical differences in the tolerance, uptake and transport capacity of Cd at different concentrations. We identified the differentially expressed gene and profiled the molecular regulatory pathways of perennial ryegrass with GO and KEGG analysis in response to Cd stress. The results show that the ryegrass response to cadmium stress involved the participation of multiple genes and the co-regulation of multiple biological processes, particularly biological regulation (GO:0065007), metabolic process (GO:0008152) and cellular process (GO:0009987). The activities of key enzymes related to the antioxidant activity (of POD, CAT and SOD, etc.) and corresponding gene expression levels were significantly changed under Cd stress. The growth physiology and antioxidant enzyme system of ryegrass were damaged by high concentrations of Cd. Studies have shown that Cd stress may cause changes in metabolites such as sugar, amino acids and organic acids. KEGG enrichment of metabolic pathways demonstrated that environmental adaption, carbohydrate metabolism, terpenoids and polyketides metabolism, amino acid metabolism, lipid metabolism, signal transduction, translation and folding, sorting and degradation, transport and catabolism were significantly changed under cadmium stress, of which up- or down-regulated genes might contribute to the enrichment of heavy metals in ryegrass. Further research is needed to identify important candidate genes and markers linked to Cd tolerance that can be applied to improve the genetics of perennial ryegrass for enhanced Cd tolerance. This study successfully demonstrates the effects of high Cd concentration on ryegrass physiology and transcriptome levels, and has important guiding significance for the prevention and control of heavy metal pollution in the future.

## 5. Materials and Methods

### 5.1. Experimental Design

The pot experiment was conducted from 12 August to 26 October 2019 in a sunshine greenhouse of the Agricultural Water and Soil Environmental Field Science Research Station, Chinese Academy of Agricultural Science, located in Henan Province of China (Xinxiang, Henan Province, in North China) (35°19′ N, 113°53′ E). The experimental soil was collected from the suburban farmland, and the soil properties were pH 8.24, electrical conductivity (EC) 439 μS·cm^−1^, organic matter (OM) 0.45%, total nitrogen (TN) 0.39 g·kg^–1^ and total phosphate (TP) 0.80 g·kg^–1^, and total cadmium (Cd), lead (Pb), copper (Cu) and zinc (Zn) levels were 0.07 mg·kg^–1^, 7.99 mg·kg^−1^, 5.18 mg·kg^–1^ and 32.16 mg·kg^–1^, respectively.

7.5 kg of air-dried soil in each pot was passed through a 20-mesh screen, and then treated with Cd levels of 0 mg·kg^–1^ (C0), 5 mg·kg^–1^ (C5), 25 mg·kg^–1^ (C25), 50 mg·kg^–1^ (C50), 100 mg·kg^−1^ (C100) and 500 mg·kg^−1^ (C500) (CdCl_2_·2.5H_2_O), and mix thoroughly. The base fertilizer included N (Urea), P (NH_4_H_2_PO_4_) and K (KCl) at concentrations of 200 mg·kg^–1^, 150 mg·kg^–1^ and 150 mg·kg^–1^, respectively. The selected plant used in the experiment was ryegrass (*Lolium perenne* L.). Ryegrass seeds were germinated in a plastics cave dish containing a common nursery substrate of grass peat, vermiculite and perlite in a 1:1:1 ratio. After one month of equilibration of the contaminated/selected soil with heavy metal solutions, ten seedlings with uniform growth potential were transplanted into each pot. The ryegrass plants were grown in a sunshine greenhouse under controlled conditions for two and a half months. All samples were kept in a portable icebox during sampling, and then transported to the laboratory and stored at 4 °C prior to processing for subsequent analyses.

### 5.2. Physicochemical Characteristics Analysis

The soil (10-15 cm) was collected and air-dried under ambient temperature, and then sieved through a 2 mm mesh for the physicochemical properties analysis according to standard methods (GB 15618-2018). Total concentrations of Cd were determined by digesting 0.3 g of each soil and plant in HF-HNO_3_-HClO_4_ in a microwave oven, and measured by atomic absorption spectrophotometer (Shimadzu AA-6300, Kyoto, Japan). Total nitrogen (TN) concentrations in the soil were measured by ultraviolet spectrophotometry with basic potassium persulfate digestion. 0.3 g of 100 mesh-sieved soil samples were weighted and digested with H_2_SO_4_-HClO_4_, then total phosphorus (TP) determined by molybdenum-antimony anti-spectrophotometry.

Plant height, fresh weight and tiller numbers of individual plants were measured manually. Aerial parts were analyzed for antioxidant enzyme activities (SOD, POD and CAT), chlorophyll a content (Chla) and lipid peroxidation (Malondialdehyde, MDA). Leaves were ground and extracted to obtain crude enzyme solution. They were then briefly rinsed with deionized water and ground. Samples (100 mg) were placed in a phosphate-buffered saline (PBS) solution (50 mM, pH 7.8), and extractants were centrifuged (8000 g) for 10 min at 4 °C to collect the crude enzyme solution. SOD, POD, CAT and MDA were measured by assay kits (Solarbio Science and Technology Co., Ltd., Beijing, China).

### 5.3. RNA Extraction

Ryegrass leaf samples were rinsed several times with deionized water, and then ground with liquid nitrogen. A 100 mg sample was used for total RNA extraction with TRNzol Universal Reagent (Tiangen Biotech, Beijing, China), following the manufacturer’s instruction. The total RNA quality was determined by a 2100 Bioanalyser (Agilent Technologies, Palo Alto, CA, USA) and quantified using the NanoDrop-2000 (Thermo Scientific, Wilmington, DE, USA). The isolated RNA was divided into two aliquots: one aliquot for RNA sequencing library construction; and another aliquot for qRT-PCR.

### 5.4. Gene Expression of Antioxidant Enzymes

RNA (1.0 μg) was reverse-transcribed with theFastKing RT Kit (With gDNase) (TiangenBiotech, Beijing, China). The primers of chloroplastic copper/zinc (Cu/ZnSOD), iron SOD (FeSOD), manganese SOD (MnSOD), CAT, POD and ascorbate peroxidase (APX) were designed by Bian et al. (2009) [42]. The reactions were performed using the TB Green™ Premix Ex Taq™ (Tli RNaseH Plus) (TaKaRa Bio Inc., Dalian, China) on a C1000 Touch thermal cycler with CFX96 Touch™ Real-Time PCR Detection System (Bio-Rad Laboratories, Inc., Hercules, CA, USA). The thermal cycling conditions used were as follows: initial denaturation at 95 °C for 15 s, 40 cycles of 95 °C for 5 s, 55 °C for 30 s and 72 °C for 30 s, followed by a melt curve stage from 65 °C, gradually increasing from 0.5 °C·s^–1^ to 95 °C.

### 5.5. Illumina Sequencing

A total 1.0 μg RNA from each group (C0, C50 and C500 samples) was used for the RNA-Seq analysis. The TruSeqTM RNA Sample Preparation Kit from Illumina (San Diego, CA, USA) was used to generate the RNA-Seq libraries according to the manufacturer’s instructions. Libraries were size selected for cDNA target fragments of 300 bp on 2% Low Range Ultra Agarose, followed by PCR amplified using Phusion DNA polymerase (NEB, Ipswich, MA, USA) for 15 PCR cycles. After being quantified by TBS380, RNA-Seq sequencing library was sequenced with the Illumina HiSeqXTen platform using paired-end reads (2 × 150 bp read length). The sequencing was carried out at Majorbio Bio-Pharm Technology Co., Ltd. (Shanghai, China).

### 5.6. Differential Expression Genes and Functional Enrichment Analysis

To identify differential expression genes (DEGs) between two different samples, the expression levels of genes were calculated according to the Fragments Per Kilobases per Million reads (FPKM) method. RSEM (http://deweylab.github.io/rsem/, accessed on 15 October 2022) was used to quantify the gene expression levels [43]. The DESeq2 was used to analyze and identify the differential expression genes [44]. Genes with P-adjust (FDR) ≤ 0.05 and a |log_2_ ratio| ≥ 1 were identified as DEGs. Additionally, functional enrichment analyses, including GO and KEGG, were performed. The GO functional enrichment and KEGG pathway analysis were carried out by Goatools (https://github.com/tanghaibao/Goatools, accessed on 15 October 2022) and KOBAS (http://kobas.cbi.pku.edu.cn/home.do, accessed on 15 October 2022) [45].

### 5.7. Quantitative Real-time PCR Validation

The relative expression levels of 13 randomly selected DEGs were validated with RT-qPCR. RT-qPCR was performed on C1000 Touch thermal cycler with CFX96 Touch™ Real-Time PCR Detection System (Bio-Rad Laboratories, Inc., Hercules, CA, USA) using TB Green Premix Ex Taq (Tli RNaseH Plus) (TaKaRa Bio Inc., Dalian, China). The reaction conditions consisted of an initial denaturing step of 95 °C for 15 s, followed by 40 cycles of 95 °C for 5 s, 60 °C for 30 s and 72 °C for 30 s. The conditions included a melting temperature of 60–65 °C. Primer sequences for RT-qPCR were designed using Primer-BLAST (https://www.ncbi.nlm.nih.gov/tools/primer-blast/index.cgi, accessed on 15 October 2022) and listed in Appendix A. The *eIF4A* was used as an internal standard. The relative gene expression levels were calculated by the 2^−ΔΔCt^ method. All tests were performed in triplicate for the reference and targeted genes [46].

### 5.8. Statistical Analysis

The results were expressed as the means ± standard deviations. All data were analyzed and statistically examined by one-way ANOVA with a Duncan’s test using SPSS version 27 (SPSS, Inc., Chicago, IL, USA). Statistical significance was considered at *p* < 0.05, unless specified otherwise.

## Figures and Tables

**Figure 1 toxics-10-00734-f001:**
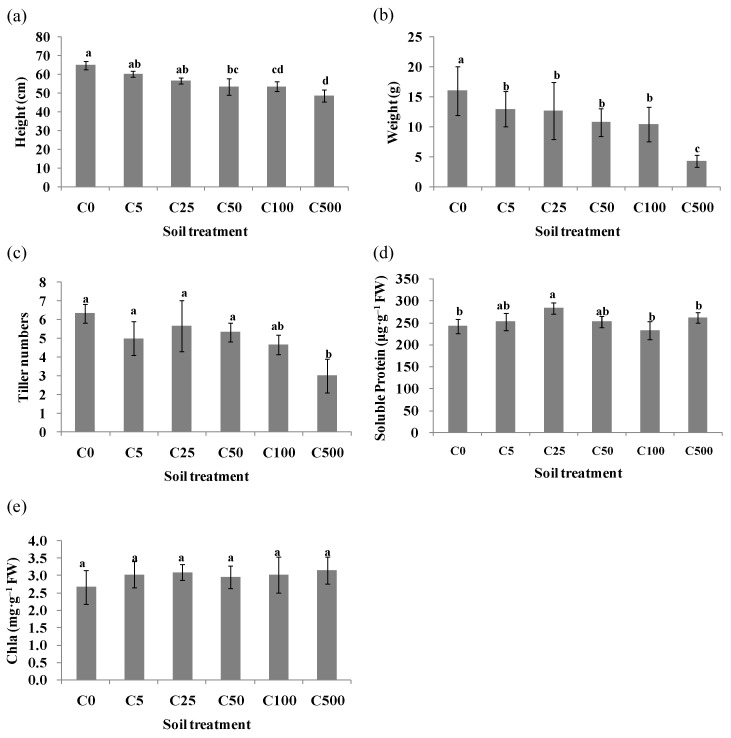
The growth physiological analysis of ryegrass under Cd stress for (**a**) height, (**b**) weight, (**c**) tiller number, (**d**) soluble protein and (**e**) chla content. Values are expressed as means of three replicates with standard deviations (±SD). Different lowercase letters indicate that values are significantly different at *p* < 0.05.

**Figure 2 toxics-10-00734-f002:**
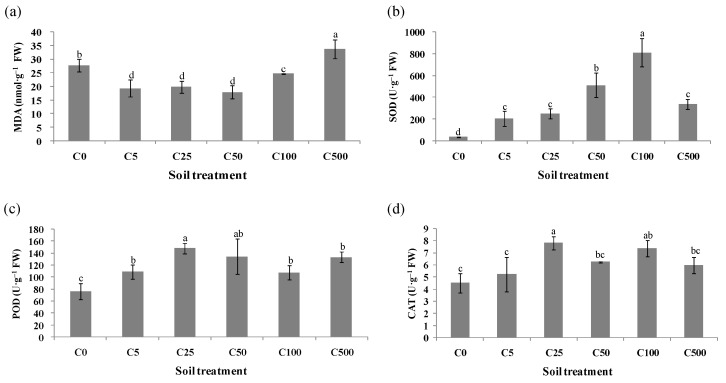
Changes of (**a**) MDA, (**b**) SOD, (**c**) POD and (**d**) CAT in ryegrass leaves under Cd stress. Values are expressed as means of three replicates with standard deviations (±SD). Different lowercase letters indicate that values are significantly different at *p* < 0.05.

**Figure 3 toxics-10-00734-f003:**
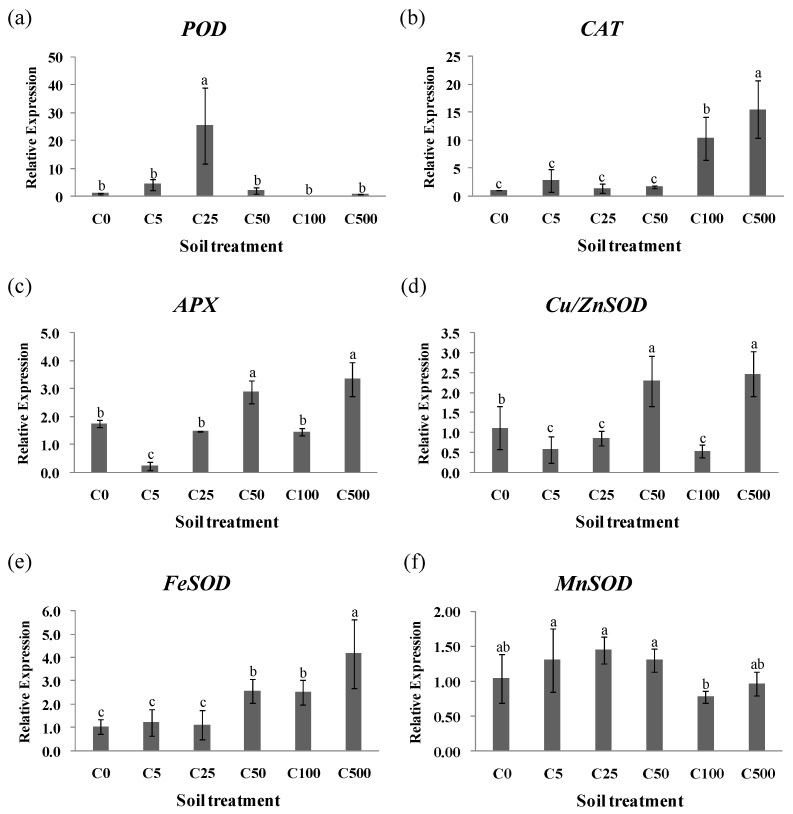
Gene expressions of (**a**) *POD*, (**b**) *CAT*, (**c**) *APX*, (**d**) *Cu/ZnSOD*, (**e**) *FeSOD* and (**f**) *MnSOD* in ryegrass leaves under Cd stress. Values are the means of three replications ±SD. Different lowercase letters indicate significant difference among treatments (*p* < 0.05).

**Figure 4 toxics-10-00734-f004:**
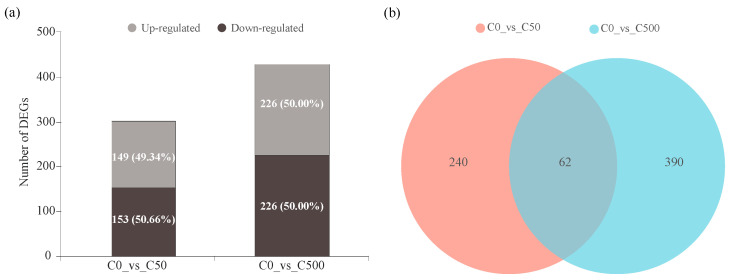
(**a**) Number of up- and down-regulated DEGs in various Cd concentrations. (**b**) Venn diagram of all DEGs. C0: group treated with 0 mg·kg^–1^ cadmium. C50: group treated with 50 mg·kg^–1^ cadmium. C500: group treated with 500 mg·kg^–1^ cadmium.

**Figure 5 toxics-10-00734-f005:**
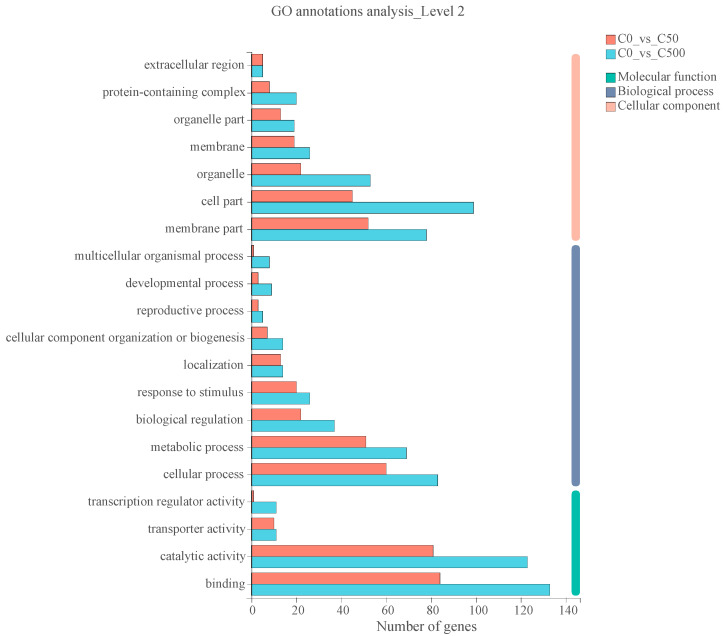
GO function classification of the specific expressed genes under Cd stress.

**Figure 6 toxics-10-00734-f006:**
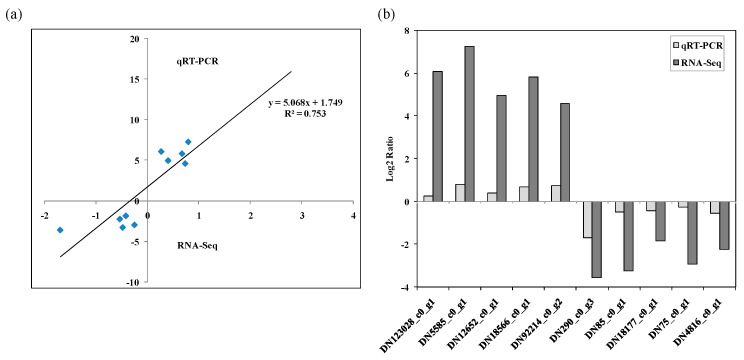
Validation of DEGs in leaves of ryegrass exposed to Cd stress. (**a**) the correlation between the RNA-Seq and qRT-PCR data from 10 selected DEGs in leaves of ryegrass exposed to Cd stress, (**b**) comparison of the relative expression of 10 selected DEGs in ryegrass leaves exposed to Cd stress by RNA-Seq and qRT-PCR.

## Data Availability

The data presented in this study are available in Appendix A (Appendix A; Appendix A).

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
