# Peer review of "Transcriptomic Sequencing Analysis on Key Genes and Pathways Regulating Cadmium (Cd) in Ryegrass (Lolium perenne L.) under Different Cadmium Concentrations"

_toxics, 2022, doi:10.3390/toxics10120734_

Round 1

Reviewer 1 Report

See attached MS word file for comments

Line 14-16: The English in the sentence is awkward and should be re-written

Line 19-21: The English in the sentence is awkward and should be re-written

Line 21: rt-qPCR Should be RT-qPCR

Line 26-28: The English in the sentence is awkward and should be re-written

Line 31-33: The English in the sentence is awkward and should be re-written

Line 43-50: The English in the paragraph is awkward and should be re-written

Line 52: What is “It” referring to? Phytoremediation? Ryegrass? Please rewrite.

Line 56: tolerant should be the word tolerance

Line 58-60: Ryegrass by accepted definition of what metal hyper accumulator plant is (0.1-10% metal content and plant still physiologically unaffected with regard to growth inhibition) should not be classified as a metal tolerant or accumulating plant.

Line 63: Aspergillus aculeatus should be in italics

Line 84: (Triticum urartu) should be in italics

Line 90 Hu et al. should have a year associated with it

On first use of an abbreviated term, it should be fully spelled out. i.e. Superoxide Dismutase (SOD)

Results section:

For the experimental design it may have been useful to study some intermediate concentrations of cadmium in the soil treatments such as 10 or 25 mg/kg soil and 250 mg/kg soil.

Figure 1: Should have a X -axis label. Maybe something like Cadmium soil treatment? Or Soil treatment?

Figure 2: Should have a X -axis label. Maybe something like Cadmium soil treatment? Or Soil treatment?

Figure 3: Should have a X -axis label. Maybe something like Cadmium soil treatment? Or Soil treatment? Not sure what the red (5.00) and the (0.50) on the POD APX and Cu/Zn SOD graphs are. Also not sure why graphs have Y-axes with tenths and hundredths of units displayed? Should only have the ones place displayed. i.e. why 20.00 and not just 20?

Figure 4: not sure if it is the draft article but the resolution of this figure it is very low and blurry.

C0 vs C50 and C0 vs C500 are done but C0 vs C5, C0 vs C25 and C0 vs C100 are missing? These should be at least commented on or addressed.

Figure 5: not sure if it is the draft article but the resolution of this figure it is very low and blurry

I would also suggest further defining what molecular function, biological process and cellular component mean and define them better.

Discussion section:

Spell out abbreviations on first use. The paper uses a lot of abbreviations and can be confusing at times.

Lines 255-299: this is one huge paragraph with many subjects addressed. This paragraph should be broken up into paragraphs for each subject.

Conclusions section:

Should be more focused and detail which genes found in the transcriptome analysis may be the most significant genes involved in the response to Cd and the overall use of Ryegrass for phytoremediation purposes.

Overall thoughts:

The science in the paper is very good. There are all the appropriate controls, and the experimental methods are sound. The most significant aspect of the paper that would most contribute to the scientific community and the body of knowledge of science is the Transcriptome analysis. This aspect of the paper should be given the highest prominence in the writing of the paper. Also, more interpretation in depth of these results would strengthen the paper greatly. My advice to the authors is to make these results more prominent in the discussion and conclusions.

Finally, have the paper edited by a native English speaker and writer would significantly improve the flow, readability, and impact of the results of the science.

Author Response

Reviewer 1

Line 14-16: The English in the sentence is awkward and should be re-written

Answer: Thanks for your comments. The English in the sentence has been rewritten in the revised manuscript.

Line 19-21: The English in the sentence is awkward and should be re-written

Answer: Thanks for your comments. The English in the sentence has been rewritten in the revised manuscript.

Line 21: rt-qPCR Should be RT-qPCR

Answer: Thanks for your comments. “rt-qPCR” was changed to “RT-qPCR”.

Line 26-28: The English in the sentence is awkward and should be re-written

Answer: Thanks for your comments. The English in the sentence has been rewritten in the revised manuscript.

Line 31-33: The English in the sentence is awkward and should be re-written

Answer: Thanks for your comments. The English in the sentence has been rewritten in the revised manuscript.

Line 43-50: The English in the paragraph is awkward and should be re-written

Answer: Thanks for your comments. The English in the paragraph has been rewritten in the revised manuscript.

Line 52: What is “It” referring to? Phytoremediation? Ryegrass? Please rewrite.

Answer: Thanks for your comments. "It" refers to plant, and has been modified in the revised manuscript.

Line 56: tolerant should be the word tolerance

Answer: Thanks for your comments. “tolerant” was corrected as “tolerance” in the revised manuscript.

Line 58-60: Ryegrass by accepted definition of what metal hyper accumulator plant is (0.1-10% metal content and plant still physiologically unaffected with regard to growth inhibition) should not be classified as a metal tolerant or accumulating plant.

Answer: Thanks for your comments. This sentence has been rephrased in the revised manuscript.

Line 63: Aspergillus aculeatus should be in italics

Answer: Thanks for your comments. Aspergillus aculeatus should be in italics, and has been deleted due to it is not relevant to the text.

Line 84: (Triticum urartu) should be in italics

Answer: Thanks for your comments. Triticum urartu has been italicized in the revised manuscript.

Line 90 Hu et al. should have a year associated with it

Answer: Thanks for your comments. Year has been added in the revised manuscript.

On first use of an abbreviated term, it should be fully spelled out. i.e. Superoxide Dismutase (SOD)

Answer: Thanks for your comments. First use of all abbreviations has been fully spelled out in the revised manuscript. i.e. superoxide dismutase (SOD), peroxidase (POD), catalase (CAT), etc.

Results section:

For the experimental design it may have been useful to study some intermediate concentrations of cadmium in the soil treatments such as 10 or 25 mg/kg soil and 250 mg/kg soil.

Answer: Thanks for the insightful concern. Indeed, we analyzed the physicochemical parameters changed under a series of concentrations of Cd exposure, as well as the transcriptional changes of genes coding for antioxidant enzymes. While to further explore the whole transcriptional changes of ryegrass under stress, we chose C50 and C500, which represent the moderately and highly contaminated conditions, respectively.

Figure 1: Should have a X -axis label. Maybe something like Cadmium soil treatment? Or Soil treatment?

Answer: Thanks for your comments. "Soil treatment" has been added on the X axis in the revised manuscript.

Figure 2: Should have a X -axis label. Maybe something like Cadmium soil treatment? Or Soil treatment?

Answer: Thanks for your comments. "Soil treatment" has been added on the X axis in the revised manuscript.

Figure 3: Should have a X -axis label. Maybe something like Cadmium soil treatment? Or Soil treatment? Not sure what the red (5.00) and the (0.50) on the POD APX and Cu/Zn SOD graphs are. Also not sure why graphs have Y-axes with tenths and hundredths of units displayed? Should only have the ones place displayed. i.e. why 20.00 and not just 20?

Answer: Thanks for your comments. "Soil treatment" has been added on the X axis in the revised manuscript. The Y-axis scale has been modified.

Figure 4: not sure if it is the draft article but the resolution of this figure it is very low and blurry.

Answer: Thanks for your comments. Figure 4 was redrawn to improve clarity in the revised manuscript.

C0 vs C50 and C0 vs C500 are done but C0 vs C5, C0 vs C25 and C0 vs C100 are missing? These should be at least commented on or addressed.

Answer: Thanks for the insightful concern. As described in the above answer, to explore the whole transcriptional changes of ryegrass under stress, we chose C50 and C500, which represent the moderately and highly contaminated conditions, respectively. As suggested, we explained this in the revised manuscript in the “material and methods” part.

Figure 5: not sure if it is the draft article but the resolution of this figure it is very low and blurry

Answer: Thanks for your comments. Figure 5 was redrawn to improve clarity in the revised manuscript.

I would also suggest further defining what molecular function, biological process and cellular component mean and define them better.

Answer: Thanks a lot for this suggestion. We do the GO analysis using GOATOOLS (Klopfenstein, et al., 2018), the GO contains three hierarchies that represent gene function based on categories of molecular function (MF), biological process (BP), and cellular component (CC). Specifically, MF refers to the action or activity performed by the gene product, BP represents a specific objective that the organism is genetically programmed to achieve and CC refers to genes relative to cellular compartments and structures. As suggested, we added this description in the revised manuscript.

References

Klopfenstein, D.V., Zhang, L., Pedersen, B.S. et al. GOATOOLS: A Python library for Gene Ontology analyses. Sci Rep 8, 10872 (2018). https://doi.org/10.1038/s41598-018-28948-z

Discussion section:

Spell out abbreviations on first use. The paper uses a lot of abbreviations and can be confusing at times.

Answer: Thanks for your comments. First use of all abbreviations has been fully spelled out in the revised manuscript.

Lines 255-299: this is one huge paragraph with many subjects addressed. This paragraph should be broken up into paragraphs for each subject.

Answer: Thanks for your comments. According to the reviewer's suggestion, this paragraph is divided into two parts in the revised manuscript based on the subjects.

Conclusions section:

Should be more focused and detail which genes found in the transcriptome analysis may be the most significant genes involved in the response to Cd and the overall use of Ryegrass for phytoremediation purposes.

Answer: Thanks for the insightful concern. As suggested, we added some detail information for the genes that may have significant functions in response to Cd exposure in the results and discussion parts.

Overall thoughts:

The science in the paper is very good. There are all the appropriate controls, and the experimental methods are sound. The most significant aspect of the paper that would most contribute to the scientific community and the body of knowledge of science is the Transcriptome analysis. This aspect of the paper should be given the highest prominence in the writing of the paper. Also, more interpretation in depth of these results would strengthen the paper greatly. My advice to the authors is to make these results more prominent in the discussion and conclusions.

Answer: Thanks for the insightful comments. As suggested, we highlight the importance of relevant results (transcriptome analysis) in the discussion and conclusions.

Finally, have the paper edited by a native English speaker and writer would significantly improve the flow, readability, and impact of the results of the science.

Answer: Thanks for your comments. The author used MDPI language editing services to polish the English to improve the readability of the paper.

Again, thanks a lot for all your distinctive comments and suggestions.

Reviewer 2 Report

This paper presents transcriptomic results on ryegrass under Cd stress. Although the novelty is not great the authors present interesting results.

The authors choose to use high Cd concentrations, the highest one (500 mg/kg) is extremely high and does not occur in nature so the authors must explain why they use the selected Cd concentrations.

Abstract:

The authors should avoid using acronyms in the abstract (like SOD etc).

Line 20: The abstract ony mentions 3 Cd concentrations but several analytical procedures were performed with more Cd concentrations.

Introduction:

Line 63: Aspergillus aculeatus should be in italics. Anyway, this sentence should be removed as this paper does no deal with fungi.

Results

Lines 106-108 must be removed.

Line 159: Why is The comparative transcriptomic analysis performed only in 3 Cd concentrations?

Figure 4 has very low quality and is almost unreadable, please replace.

Figure S4 has no text in it, as is its difficult to understand it.

Discussion:

Remove lines 251-254.

Generally the authors should mention that 500 mg/kg (even 50!) is an extremely high Cd concentration, and thus effects are expected, but that have little relation to what happens in real world Cd-contaminated soils

Line 286: Correct the sentence: “Noteworthy, there had enormous implications…”

Line 308: This sentence should be rewritten as “Metallothioneins (MTs) are small metal binding proteins that can bind…”

Line 317: This sentence should be rewritten as  “Phytochelatins (PCs) can chelate Cd2+, Zn2+ and Cu2+ entering plant cells…”

Line 322: Please correct the name Dominguez-Solis.

Materials and methods:

Why was the transcritmoic analysis made only in 3 Cd concentrations? This must be explained.

The authors do not describe the experimental methods, for example, the soil and plant analysis for Cd is only generally mentioned. This can be accepted but only if they make reference to a published paper where that method was described.

Line 400: “-1” should be in subscript, make sure its correctly written everywhere in the text.

Author Response

Reviewer 2

This paper presents transcriptomic results on ryegrass under Cd stress. Although the novelty is not great the authors present interesting results.

The authors choose to use high Cd concentrations, the highest one (500 mg/kg) is extremely high and does not occur in nature so the authors must explain why they use the selected Cd concentrations.

Answer: Thanks for your comments. As reported in the references, Cd contents in polluted soils showed a wide range, with maximum concentrations over hundred mg/kg (Zhang et al., 2015, Kubier et al., 2019), therefore the 500 mg/kg condition represent a highly contaminated condition, and we thus analyzed the transcriptional changes in C500.

Abstract:

The authors should avoid using acronyms in the abstract (like SOD etc).

Line 20: The abstract only mentions 3 Cd concentrations but several analytical procedures were performed with more Cd concentrations.

Answer: Thanks for the insightful concern. As suggested, we added the results of the other Cd concentrations.

Introduction:

Line 63: Aspergillus aculeatus should be in italics. Anyway, this sentence should be removed as this paper does no deal with fungi.

Answer: Thanks for your comments. Aspergillus aculeatus should be in italics, this sentence has been deleted due to it does not relevant to the paper.

Results

Lines 106-108 must be removed.

Answer: Thanks for your comments. The sentences have been deleted.

Line 159: Why is The comparative transcriptomic analysis performed only in 3 Cd concentrations?

Answer: Thanks a lot for the question. Since our research purpose was to explore the responses of ryegrass under Cd stress. As shown in Figure 1-2, compared to C5 and C25, parameters including height, weight, tiller numbers, and the contents of MDA, POD showed no statistical significance in C50 samples, therefore, based on these physicochemical parameters changed under Cd stress, we chose to analyzed the transcriptional changes of samples in C50, to represent a moderately contaminated condition. Moreover, as reported in the references, Cd contents in polluted soils showed a wide range, with maximum concentrations over hundred mg/kg (Zhang et al., 2015, Kubier et al., 2019), therefore the 500 mg/kg condition represent a highly contaminated condition, and we thus analyzed the transcriptional changes in C500. In addition, we list some references that focus on the scope of “Cd and ryegrass” in the following table, the concentrations of Cd in these references are varied based on their experiment purposes.

Purpose

Culture

Cd concentrations

References

Study the Cd tolerance of ryegrass

Soilless culture

50 - 1000 μM (5.6 -112 mg/kg)

(Fang et al., 2017)

Study the phytoextraction efficiency of ryegrass

In soil

10 - 50 mg/kg

(Zhang et al., 2019)

Study the expression of MT (sulfhydryl on metallothioneins) genes in ryegrass

In soil

75 - 600 mg/kg

(Li et al., 2019)

Cadmium uptaken by ryegrass

In soil

16.8 mg/kg

(Li et al., 2020, Li et al., 2021)

Cadmium uptaken by ryegrass

In soil

3.68 mg/kg

(Benyas et al., 2018)

Study the transcriptional responses in roots under Cd stress

Soilless culture

0.5 - 3.5 mM (56 - 392 mg/kg)

(Wang et al., 2020)

Reference

Benyas E., Owens J., Seyedalikhani S., Robinson B., 2018. Cadmium uptake by ryegrass and ryegrass-clover mixtures under different liming rates. J. Environ. Qual., 47 (5), 1249-1257.

Fang Z., Hu Z., Zhao H., Yang L., Ding C., Lou L., et al., 2017. Screening for cadmium tolerance of 21 cultivars from Italian ryegrass (Lolium multiflorum Lam.) during germination. Grassl. Sci., 63 (1), 36-45.

Kubier A., Wilkin R.T., Pichler T., 2019. Cadmium in soils and groundwater: A review. Appl. Geochem., 108, 1-16.

Li G., Chen F., Jia S., Wang Z., Zuo Q., He H., 2020. Effect of biochar on Cd and pyrene removal and bacteria communities variations in soils with culturing ryegrass (Lolium perenne L.). Environ. Pollut., 265 (Pt A), 114887.

Li G.R., Wang Z.S., Lv Y.J., Jia S.Y., Chen F.K., Liu Y.B., et al., 2021. Effect of culturing ryegrass (Lolium perenne L.) on Cd and pyrene removal and bacteria variations in co-contaminated soil. Environmental Technology & Innovation, 24.

Li Y.H., Qin Y.L., Xu W.H., Chai Y.R., Li T., Zhang C.L., et al., 2019. Differences of Cd uptake and expression of MT family genes and NRAMP2 in two varieties of ryegrasses. Environ. Sci. Pollut. Res. Int., 26 (14), 13738-13745.

Zhang J., Yang N., Geng Y., Zhou J., Lei J., 2019. Effects of the combined pollution of cadmium, lead and zinc on the phytoextraction efficiency of ryegrass (Lolium perenne L.). RSC Advances, 9 (36), 20603-20611.

Zhang X., Chen D., Zhong T., Zhang X., Cheng M., Li X., 2015. Assessment of cadmium (Cd) concentration in arable soil in China. Environ. Sci. Pollut. Res. Int., 22 (7), 4932-4941.

Wang J., Zhao J., Feng S., Zhang J., Gong S., Qiao K., et al., 2020. Comparison of cadmium uptake and transcriptional responses in roots reveal key transcripts from high and low-cadmium tolerance ryegrass cultivars. Ecotoxicol. Environ. Saf., 203, 110961.

Figure 4 has very low quality and is almost unreadable, please replace.

Answer: Thanks for your comments. Figure 4 was redrawn to improve clarity in the revised manuscript.

Figure S4 has no text in it, as is its difficult to understand it.

Answer: Thanks for your comments. Figure S4 was redrawn and annotated.

Discussion:

Remove lines 251-254.

Answer: Thanks for your comments. The sentences have been deleted.

Generally the authors should mention that 500 mg/kg (even 50!) is an extremely high Cd concentration, and thus effects are expected, but that have little relation to what happens in real world Cd-contaminated soils

Answer: Thanks for the insightful concern. As reported in the published studies, Cd contents in polluted soils showed a wide range, with maximum concentrations over hundred mg/kg (Zhang et al., 2015, Kubier et al., 2019), so we set 500 mg/kg, to represent a highly contaminated condition.

References

Kubier A., Wilkin R.T., Pichler T., 2019. Cadmium in soils and groundwater: A review. Appl. Geochem., 108, 1-16.

Zhang X., Chen D., Zhong T., Zhang X., Cheng M., Li X., 2015. Assessment of cadmium (Cd) concentration in arable soil in China. Environ. Sci. Pollut. Res. Int., 22 (7), 4932-4941.

Line 286: Correct the sentence: “Noteworthy, there had enormous implications…”

Answer: Thanks for your comments. This sentence has been corrected in the revised manuscript.

Line 308: This sentence should be rewritten as “Metallothioneins (MTs) are small metal binding proteins that can bind…”

Answer: Thanks for your comments. This sentence has been rewritten in the revised manuscript.

Line 317: This sentence should be rewritten as  “Phytochelatins (PCs) can chelate Cd2+, Zn2+ and Cu2+ entering plant cells…”

Answer: Thanks for your comments. This sentence has been rewritten in the revised manuscript.

Line 322: Please correct the name Dominguez-Solis.

 Answer: Thanks for your comments. "Dominguez-Solis" was corrected as “Domínguez-Solís” in the revised manuscript.

Materials and methods:

Why was the transcritmoic analysis made only in 3 Cd concentrations? This must be explained.

Answer: Thanks a lot for the question. As described in the above comment, based on the physicochemical parameters changed under Cd stress, and the Cd contents in polluted soils reported in published papers, we chose to analyze the transcriptional changes of C50 and C500, representing a moderately and a highly contaminated condition, respectively.

The authors do not describe the experimental methods, for example, the soil and plant analysis for Cd is only generally mentioned. This can be accepted but only if they make reference to a published paper where that method was described.

Answer: Thanks for your comments. The experimental methods have been modified in the revised manuscript.

Line 400: “-1” should be in subscript, make sure its correctly written everywhere in the text.

Answer: Thanks for your comments. All "-1" that appear in the text have been changed to superscript.

Again, thanks a lot for all your distinctive comments and suggestions.
